# The Effect of Digital Transformation on the Pharmaceutical Sustainable Supply Chain Performance: The Mediating Role of Information Sharing and Traceability Using Structural Equation Modeling

**Jing-Yan Ma [1], Lei Shi [2] and Tae-Won Kang [1,\*]**

[1] Department of Supply Chain and Logistics, College of Social Sciences, Kunsan National University, Gunsan 54150, Republic of Korea
[2] Business College, Wuxi Taihu University, Wuxi 214000, China
\* Correspondence: twkang@kunsan.ac.kr

**Abstract:** As the global pharmaceutical market continues to expand, the demand for pharmaceutical supply chain is increasing. In the context of "Industry 4.0", the pharmaceutical supply chain sector needs to accelerate digital construction. Pharmaceutical companies need to strengthen risk management in order to cope with supply disruptions. From the perspective of sustainable development, the pharmaceutical supply chain can achieve sustainable supply performance in social, economic and environmental dimensions through digital transformation. There is a lack of research on digital transformation of pharmaceutical supply chain management. Further research is needed on what specific digital management pharmaceutical companies need to enhance to improve supply performance. This study uses empirical analysis to examine the impact of digital transformation on sustainable supply chain performance and to explore the role of information sharing and traceability as mediators. The aim is to guide the pharmaceutical supply chain to clearly manage the development of digital transformation and obtain sustainable supply performance. This study presents hypotheses based on cutting-edge theoretical findings. In total, 298 Chinese pharmaceutical company supply chain managers were surveyed and Structural equation analysis was conducted using SPSS26.0 and AMOS24.0. The results show that digital transformation significantly and positively impacts sustainable supply chain performance. Traceability plays a mediating role. The mediating role of information sharing is not significant. However, information sharing and traceability as two separate trends can have synergistic effects that together affect sustainable supply performance. The conclusion is that the pharmaceutical supply chain should accelerate digital construction, eliminate the uneven development of digital technology among supply chain members, and reduce the impact of technological uncertainty on performance. Companies are enhancing supply chain security management through information sharing and traceability systems, and are continuously focusing on the role of digital transformation as a driver for sustainable development.

**Keywords:** digital transformation; information sharing; traceability; sustainable supply chain performance

## 1. Introduction

The topic of sustainable development is of great interest to society today, as well as medical and healthcare. As the global population ages [1], health literacy increases and biomedical technology advances at a rapid pace, the pharmaceutical market is growing [2,3]. According to the Report of Healthcare in 2022, global per capita healthcare expenditure has continued to increase over the last five years as Figure 1 [4].

Pharmaceutical companies are focusing on how to achieve significant performance in the medical market rapid growth period. In addition, the increasing awareness of social

responsibility and environmental protection has led to a shift away from financial performance and a focus on social and environmental performance [5]. Large pharmaceutical companies are focusing more on the development of the entire supply chain and on achieving sustainable supply performance through effective supply chain management, thus ensuring a long-term competitive advantage for the company. The demand for healthcare markets will continue to grow in the future and efficient supply chain management will be a key driver to support market development and corporate innovation [6].

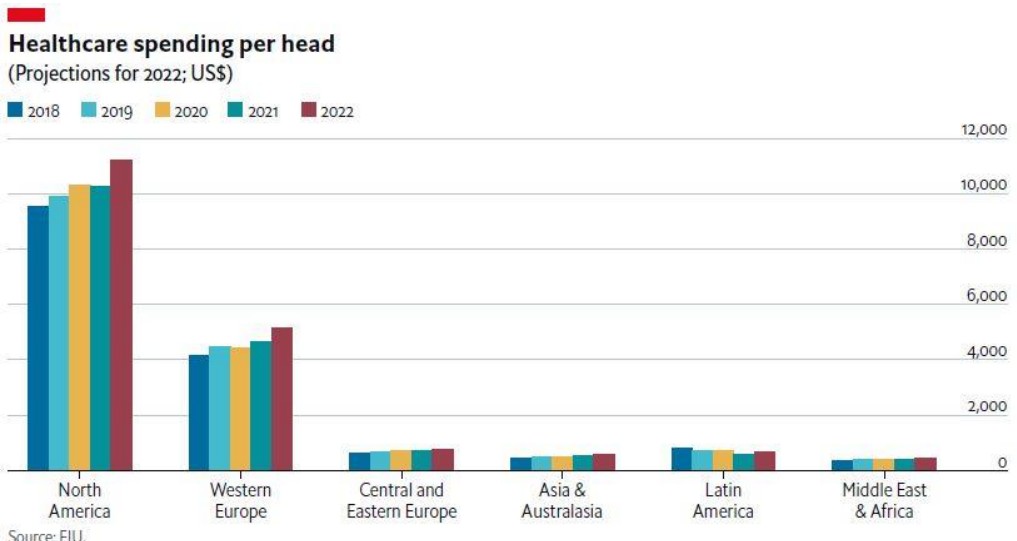

**Figure 1.** Healthcare spending per head.

The implementation of digital technologies in pharmaceutical supply chain management in the context of "Industry 4.0" makes smart manufacturing a reality. Digital technologies such as information and communication technology (ICT), cyber physical systems (CPS), the Internet of Things (IoT), blockchain, big data analytics, cloud computing, augmented reality, 3D printing, artificial intelligence, robotics and electronic data interchange (EDI) are the driving force behind 'smart manufacturing' [7]. Suppliers, manufacturers, wholesalers, retailers and end customers are connected through data and information interactions. Sharing and collaboration enable complementary information and resources across supply chain clusters, enabling agile responses to external changes and flexible strategic decisions, thereby enhancing the cluster's ability to sustain innovation [8]. Therefore, it has been suggested that information sharing, supported by digital transformation, is a catalyst for improving supply performance and continues to drive business development [9]. The manufacturing, supply and distribution processes of pharmaceuticals are more complex due to their unique nature. Therefore, traceability is particularly important in supply chain management with its ability to track and trace information [10,11]. As the application of blockchain technology penetrates deeper into the pharmaceutical supply system, the flow of products through the life cycle is precisely tracked to ensure that information is generated, aggregated, transformed and disseminated securely and efficiently [12]. Lessons from COVID-19 suggest the importance of traceable pharmaceutical supply chain management. A typical case is the availability of the COVID-19 vaccine during the outbreak. Pharmaceutical companies needed to produce large quantities in a short period of time, and to distribute and allocate them appropriately. The supply process also poses challenges for cold chain transport systems. For example, Pfizer/BioNTech vaccines are stored at temperatures between −80 °C and −60 °C and Moderna vaccines are stored at temperatures between −25 °C and −15 °C. Improper warm storage can lead to vaccine inactivation. Appropriate monitoring, regulation and tracking systems can ensure safe vaccine transfer [13]. In the future, with the rapid development of patient-centered medical technology, precise forward and reverse logistics will not be possible without the support of supply chain management

traceability. Therefore, Vishwakarma et al. 2022 suggested that digital transformation will add impetus to innovation in the pharmaceutical supply sector in terms of business models, product processes and organizational structures, and that traceability will support the pharmaceutical supply chain to achieve sustainable supply performance [14].

Ding 2018 reviews the impact of Pharma Industry 4.0 on supply chain management and suggests that the development of emerging technologies facilitates the creation of sustainable value and helps pharmaceutical companies gain a competitive advantage [15]. Burin et al. 2020 used a hierarchical regression methodology to confirm that information technology (IT) helps suppliers and distributors to complement missing information on planning and demand, thereby improving the supply chain's ability to react and adapt to environmental changes. This supply chain flexibility is an important weapon for firms to gain a competitive advantage in a dynamic environment [16]. Kavita et al. 2019 conducted an empirical study on the implementation of ICT in supply chain management in Indian pharmaceutical companies. The results of the study showed that ICT can increase the competitiveness of pharmaceutical companies and improve the information exchange between the company and its customers [17]. Alharthi et al. 2020 study the pharmaceutical supply chain in the Kingdom of Saudi Arabia (KSA) and show that blockchain is a distributed digital ledger technology that ensures transparency, traceability and security, and propose a theoretical framework for blockchain to influence the sustainability and effectiveness of the supply chain [18]. While digital transformation offers efficiency, innovation and competitiveness in the supply chain, it can also place a financial burden on pharmaceutical companies. Due to the inconsistent digitalization of supply chain member organizations, there are barriers to connectivity, inefficient coordination, etc., which affect production and reduce supply performance [19]. This study therefore seeks to address the following questions through empirical research.

**RQ1:** *Can effective digital transformation lead to sustainable supply chain performance?*

**RQ2:** *Can sustainable supply performance be improved through information sharing and traceability in supply chain management?*

**RQ3:** *Will distal effect through information sharing and traceability play a more important mediating role?*

Salehi et al. 2020 studied the Iranian veterinary pharmaceutical supply chain as the subject. Data envelopment analysis (DEA) and fuzzy data envelopment analysis (FDEA) methods were used to confirm that data-driven resilience can help the supply chain maintain robustness and optimize pharmaceutical supply chain performance [20]. Salehi et al. 2020 an empirical analysis was conducted with supply chain practitioners in the logistics, pharmaceutical, and food and beverage industries in Indonesia. The SEM analysis results concluded that adequate ICT readiness and information sharing would improve supply chain performance [21]. The innovation of this paper is to conduct an empirical analysis using structural equation modeling with Chinese pharmaceutical supply chain managers as the respondents and to combine and compare the relative roles of information sharing and traceability in pharmaceutical supply chain management to explore the relationship between digital transformation, information sharing, traceability, and sustainable supply chain performance in depth and comprehensively. The purpose of the paper is to guide pharmaceutical companies in identifying directions for improving supply management, controlling risk, accelerating sustained and robust growth in supply performance, and contributing to steady innovation in the healthcare sector. The research framework of this paper is as follows: the second section reviews the existing literature; the third section describes the research hypothesis and research methodology; and the fourth section reports the results of the data analysis. The final section summarizes the research results and revelation.

## 2. Literature Review

In order to identify the specific drivers of supply chain management in achieving sustainable supply performance, this study used the visualization software CiteSpace to conduct a scientometric analysis. The study was searched in the "Science Direct" database, a collection of academic journals in the fields of science, technology and medicine, using the keyword "Sustainable supply chain performance in the pharmaceutical supply chain". A total of 410 research articles from 2020 to 2023 were retrieved and identified. innovation", "COVID-19", "blockchain", "digitalization", "supply chain". The results of keyword information clustering provide a clear understanding of the existing research dynamics in the field of pharmaceutical supply. Combining the analysis results and research background, the existing research results are sorted out. It is found that the current research in the field of pharmaceutical supply chain management tends to innovate the supply chain through digital technology [22]. The supply chain has experienced the impact of COVID-19, and pharmaceutical companies are placing greater emphasis on risk management to avoid disruptions and ensure continuous supply [23]. From the perspective of dynamic management capability theory (dynamic management capability is the ability that managers use to structure, integrate and reconfigure organizational resources). Pharmaceutical companies improve the ability of information sharing and traceability performance of the supply chain by introducing digital technologies such as blockchain to effectively improve supply continuity [24]. In the future, the supply chain will be promoted to achieve performance excellence from a sustainable development perspective that includes social, economic and environmental aspects [25].

### 2.1. Digital Transformation

Digital transformation (DT) refers to the changes in organizations caused by digital technologies, which transform the organization by integrating digital technologies and business processes. Nowadays, DT is considered to be the driver of change in the business environment. From a dynamic capability perspective, DT fundamentally improves business performance by aligning strategic direction, marketing, consumer behavior, and supply chain management [26,27]. The key role of DT in the development of sustainable supply chain strategies is highlighted by Stroumpoulis and Kopanaki (2022) through a review of the literature on supply chain management. Digital technologies are integrated in all parts of the organization to improve business processes, customer relationships and operational performance [28]. In the biopharmaceutical industry, for example, big data analytics capabilities shorten new drug development cycles, artificial intelligence streamlines the clinical trial process, robotics reduces production costs and blockchain technology enhances drug safety monitoring. The use of digital transformation improves product lifecycle management and enhances the ability of companies to innovate openly [29,30]. It is also driving a paradigm shift in the healthcare industry. Digital solutions for consumer engagement are more market competitive. E-commerce is a typical example of digital technology integration in the retail sector. E-commerce in medicine is completed by online consultation, online purchase of medicines and online payment. Consumer satisfaction is enhanced by providing convenient, efficient and cost-effective services [31,32]. Digital transformation is changing the traditional pharmaceutical supply model, creating greater profit margins for companies and giving them an edge in a highly competitive market. Although theoretical studies indicate that digital transformation can drive supply chain development, relevant empirical studies are still insufficient, especially for the actual situation of China's pharmaceutical supply chain, which needs further examination.

### 2.2. Information Sharing

Information sharing in the supply chain mostly refers to the use of information and communication technologies to enable the exchange of information data among organizations. The digital technology dimension requires symmetrical information technology systems among organizations to enable the interoperability of data reception and exchange.

This includes technological frameworks such as distributed ledgers, big data analytics, the Internet of Things, digital twins, and electronic data interchange (EDI). Information sharing increases transparency among organizations in the supply chain [33]. Avoiding information silos can effectively solve the problem of information asymmetry hindering the development of enterprises [34]. The effective capture, integration and consolidation of information support the decision-making of the supply chain management team. Accurate strategic decisions are the basis for ensuring a company's competitive advantage [35]. By sharing orders, inventory and sales status, the uncertainty of product in terms of quality, quantity and timing is effectively reduced [36]. Companies can capture market demand, shorten delivery cycles, avoid overproduction, reduce the burden of recycling, lower supply costs and increase consumer satisfaction [37]. To achieve uniform economic performance goals, information sharing can facilitate close collaboration between supply chain members. Manufacturers, transporters, retailers and customers who benefit from information sharing build stable and reliable trust relationships with each other [38]. Effective information sharing is even more important due to the complexity of the pharmaceutical supply chain in the healthcare sector, from both an economic and social perspective, confirms that effective information sharing among organizations can contribute to strategy development and capital planning [39]. Determinants of this include: standardized information, clear instructions, and simple and clear processes. In the future, with the rise of internet-based healthcare and precision medicine, there will be a paradigm shift in the delivery of healthcare, such as telecare and mass customization of medicines, which will require effective information coordination and integration [40]. In the competitive healthcare market, it is important to make full use of information sharing systems to accurately capture information and act with agility in decision making in order to achieve sustainable closed-loop supply. Although, theoretical studies show that information sharing plays an important role in supply chain management. However, due to the unique nature of the healthcare field, it is not only related to interests but also involves more privacy issues. Therefore, due to the potential risks of information sharing, it needs to be further evaluated that the advantages of information sharing outweigh the disadvantages in the field of pharmaceutical supply.

*2.3. Traceability*

Supply chain traceability refers to the ability to track the flow of products throughout the supply chain [41]. Early traceability was limited to the ability to record the history, status, and location of an entity's identification to achieve traceability [42]. Digital technologies were mainly implemented through barcodes and radio frequency identification (RFID). Today, in order to meet the requirements for traceability accuracy, the disconnection between information flow and logistics is minimized, and errors in information dissemination and access are reduced. Emerging technologies such as blockchain, the Internet of Things (IoT) and information and communication technology (ICT) are gradually being introduced into supply chain management [11]. Full chain monitoring and management is accomplished through real-time data capture. In addition to passive traceability, big data and intelligent analysis can also be used to achieve the conversion of active traceability for risk warning and process optimization [28]. In the pharmaceutical supply sector, traceability systems improve the visibility of pharmaceutical products from raw material procurement, production and processing, transport and storage, and distribution and retail processes [43]. Some drugs such as biologics and blood products are subject to demanding transport and storage environments, and face inactivation if their ingredients are inactivated due to inadequate environmental controls [44]. The IoT system has automated intelligent reading devices that can accurately track the temperature, humidity and smoothness of the environment in which the drug is stored. In turn, real-time adjustments can be made based on changes in data to ensure the quality and safety of medicines during the supply process [45]. By integrating with blockchain technology, the data captured by sensors at a large number of nodes throughout the drug production process is stored in a distributed ledger [46]. With its decentralized technical features, it improves interoperability between industry

stakeholders, minimizes system latency and maximizes resource utilization [47]. It is also extremely difficult to tamper with recorded data due to the serialized encrypted structure (hashes) storage, effectively preventing counterfeit medicines from entering the supply market [48]. Traceability systems can also effectively respond to the deployment of medical supplies caused by unexpected events in the pharmaceutical and health sector, increasing the flexibility and agility of the pharmaceutical supply chain. Traceability plays an important role in the recall of medicines [10]. Despite the increasing import of traceability systems in supply chain management, the medical field is mostly involved in sensitive issues and the application of traceability systems is still insufficient, and the importance of traceability in supply chain management needs to be further confirmed.

*2.4. Sustainable Supply Chain Performance*

Sustainable performance of the supply chain is the performance obtained through sustainable development [49]. Mangla et al. 2020, in their study "Operational Excellence for Sustainable Supply Chain Performance", explains that evaluating supply chain performance is no longer limited to the traditional efficiency, effectiveness, and economy. Growing pressures from governments and stakeholders, supply chains need to integrate ecological, economic, and social sustainability metrics simultaneously to assess performance [25]. Govindan et al. 2013 presented a fuzzy multi-criteria approach to examine the sustainability of supply chain performance in terms of the triple bottom line (TBL) of business concepts, social, economic and environmental dimensions [50]. Slaper & Hall 2011 articulated the definition of the triple bottom line, TBL, as an accounting framework that incorporates three performance dimensions: social, environmental, and financial [51]. From a dynamic capability perspective, the ways to achieve sustainable performance can be divided into supply chain risk management and digital network capability management [52]. Supply chain disruptions can lead to significant losses in supply performance, so supply chain risk management is fundamental to achieving sustainable supply performance. The key to risk management is the construction of resilience capabilities. Supply chain resilience refers to the ability of a supply chain to respond to, withstand and recover from unexpected disruptive events to a more optimal state [53]. In a turbulent market environment, resilience can address supply chain vulnerabilities and strengthen supply chain robustness. Supply chain visibility, flexibility, adaptability and agility are the key drivers that support the four stages of supply chain resilience preparation, response and recovery [54]. This is to ensure that the complex pharmaceutical supply chain remains stable in the face of risky shocks [55]. Digitization aims to promote the integration of the digital economy with the real economy to achieve reduced operating costs, innovative products and services, increased revenue, and improved operational efficiency. Therefore, digital networking can help supply chains achieve competitive advantage [56]. Digital network construction can help supply chains achieve competitive advantage, which helps companies gain agile insight into market changes, quickly analyze decisions, takes flexible measures, ensures secure supply, and uses the collaborative capabilities of inter-organizational networks to respond to market demand and adjust the structure, speed, capacity, and delivery of supply. On the basis of providing supply and demand that matches the organization, it develops new markets for companies, expands business revenue, and adds momentum to obtain sustainable economic performance [57]. Strict network regulation also provides the security and integrity of the drug supply chain, ensuring drug quality and safety, avoiding the danger of counterfeit drugs, protecting medical privacy, and finally ensuring the healthy and stable development of the pharmaceutical supply market from a social dimension [58]. The recall process of the supply system can avoid environmental pollution caused by waste drugs, minimize the waste of resources caused by defective drugs, and ensure sustainable supply chain performance from the environmental dimension [59]. Although triple bottom line (TBL)-based sustainability is a popular recent research, the conservative and cautious attitude of society and business towards the healthcare sector has led to the lagging progress in pharmaceutical supply chain sustainability compared to other industries.

Therefore, further research is needed to confirm the drivers of sustainable performance in pharmaceutical supply chains.

## 3. Hypotheses Development

### 3.1. Research Hypothesis

#### 3.1.1. Digital Transformation and Sustainable Supply Chain Performance

Digital transformation transforms port-to-port processes in pharmaceutical supply into more efficient systems that aid pharmaceutical supply chain management, enabling on-demand procurement, flexible manufacturing and rapid distribution. Big data analysis helps suppliers identify downstream customer needs, minimize waste and maximize economics for on-demand allocation. Additive manufacturing/3D printing technologies can quickly respond to drug shortages and ensure supply continuity [19]. Artificial intelligence (AI) technology helps pharmaceutical companies shorten the clinical trial cycle for new drugs and improve drug production efficiency [60]. Smart sensor devices can track product shipments, adjust supply solutions in real time, enhance supply security and reduce cost consumption. In this process IoT, blockchain and information technology provide the basis for efficient data collection flows [56]. In total, administrators use cloud computing to assist in decision-making, risk avoidance, performance stabilization and innovation, giving companies a competitive advantage and achieving sustained growth in supply performance [61]. In terms of environmental protection, digital transformation permeates the process of procurement, manufacturing, packaging and recycling to achieve green and sustainable supply performance by reducing resource and energy consumption [62]. The digital transformation in the pharmaceutical sector is changing the health service delivery model, with new online health approaches bringing more convenience to consumers and continuing to drive the social health industry [63]. Therefore, this study proposes the following hypothesis.

**H1.** *Digital transformation positively affects sustainable supply chain performance.*

#### 3.1.2. Digital Transformation and Information Sharing

Digital transformation affects the information processing capabilities supported by digital technology [64]. Digital technology is the basis for supporting the identification, collection and processing of large amounts of information and is a prerequisite for information interaction across the system [65]. Data generated, transmitted, stored and analyzed in the digital ecosystem needs to be communicated seamlessly on the same backbone, information is refined by supply chain stakeholder organizations to achieve the same goal of maximizing benefits, and managers make the best decisions by weighing up key information [66]. The connectivity of the physical world to networked systems and the connectivity between network ports between groups is critical to the agility of information transfer [65]. Digital transformation improves the supply chain's ability to process information and increase the interoperability of information between organizations. Additionally, company management systems must have a high level of security along with compatibility [67]. This is particularly true in the pharmaceutical sector, which is highly sensitive to privacy and security issues. Blockchain's smart contract technology, with its key function feature, guarantees data reliability while ensuring information transparency throughout the chain. It promotes the level of mutual trust between partners, which in turn facilitates the full sharing of information [68,69]. Therefore, the following hypothesis is proposed in this study.

**H2.** *Digital transformation positively affects information sharing.*

#### 3.1.3. Digital Transformation and Traceability

Digital transformation aims to improve the entity through a combination of data, computing, and communication [70]. Along with the penetration of digital technology, the idea that traceability is guided by technology is proposed and how information technology

is deployed into the supply chain to enable traceability will revolve around enhancing the traceability of information flows [71]. The information being traced in the pharmaceutical supply chain includes product attributes, process attributes, logistics information, marketing attributes, participant attributes [72]. Traceability technology provides a unique identity code for each drug, monitors product status through data, and provides supply chain members with simplified, detailed, secure and reliable traceability information to automate information process management [26]. Therefore, the following hypotheses are proposed in this study.

**H3.** *Digital transformation positively affects traceability.*

### 3.1.4. Information Sharing and Sustainable Supply Chain Performance

There are cooperative as well as competitive relationships between supply chain members. It is necessary to examine whether information sharing under both conditions has a sustained positive impact on supply performance. The value of information sharing to the supply chain is divided into direct, competitive and spillover effects. In a cooperative relationship, information sharing helps to match supply and demand, mitigate the bullwhip effect and reduce manufacturing costs, a beneficial direct effect [73]. In highly competitive relationships, information sharing can lead to an increased risk of lower profits. However, when investment in services is more efficient, consumers will be more concerned about the quality of services, and information sharing will have a positive impact on economic efficiency. In the pharmaceutical supply sector, where more than one supply chain exists, information sharing in one supply chain will influence other competing chains to adjust their decisions in response to changes in market demand, a spillover effect that benefits the overall supply market development [74]. Therefore, this study proposes the following hypothesis.

**H4.** *Information sharing positively affects sustainable supply chain performance.*

### 3.1.5. Traceability and Sustainable Supply Chain Performance

Due to the complexity of the pharmaceutical supply chain, which leads to increased vulnerability, traceability helps to increase the resilience of the supply chain and assists it in coping with shocks from both external and internal sources [75]. As the problem of drug supply disruption often occurs, and the traceability system can monitor and warn in real time before a crisis occurs by monitoring the whole life cycle of drugs; when a crisis occurs, timely dispatching measures are taken to ensure the integrity of the supply chain [76]. Traceability in supply chain management ensures that the right product is delivered in the right quantity at the right time and is an essential element in building an efficient supply chain. Traceability makes product information more transparent, secure, controllable and auditable between stakeholders. Improved information quality contributes to sustainable supply chain performance [77]. Therefore, this study proposes the following hypothesis.

**H5.** *Traceability positively affects sustainable supply chain performance.*

### 3.1.6. Information Sharing and Traceability

Information sharing is a prerequisite for ensuring traceability of supply [78]. Supply chain traceability is the ability to identify and trace the history, distribution and location of products, materials and services [43]. Additionally, information sharing can assist the traceability system to perform its task more comprehensively [78]. The information shared can be a record of the pharmaceutical production process, which helps to monitor product quality and facilitate accountability for non-conforming products [79]. A grasp of the production process also facilitates downstream companies to keep track of supply quantities and facilitate accurate submission of re-production requirements [80]. The information shared can also be the status of product transportation, and by sharing the location of the product, distributors can plan product marketing. Additionally, the sharing of temperature, humidity and speed during transport helps downstream companies to

prepare delivery storage in advance [34]. The shared information can easily be product sales, which helps upstream and downstream companies to forecast changes in market demand and control purchasing quantities. Additionally, consumers' feedback on product use and demand for new products can be quickly known through the shared system for enterprise research and development, speeding up enterprise innovation. By sharing product data information, traceable product lifecycle management is achieved [81,82]. Therefore, the following hypothesis is proposed in this study.

**H6.** *Information sharing positively affects traceability.*

### 3.1.7. Mediating Role of Information Sharing

Opportunism in economics has described how information asymmetries will create room for profit, yet this low transparency and highly controlled information flow can reduce trust between members within a cluster [14]. Today, in order to adapt to market changes and maintain a competitive advantage in the digital age, companies are focusing on information integration to achieve innovative developments in the industrial sector in a collective and collaborative manner [83]. With the agglomeration effect, the value of information contributed by individual members is amplified, accelerating internal integration and external adaptation to achieve market agility. In order to access the latest data, companies often build up their digital transformation in order to flexibly adapt their market strategies [84]. Supply chain members that profit from information exchange will contribute more information out of trust and profit-driven, jointly driving the accelerated development of the industrial sector [64]. Therefore, the following hypothesis is proposed in this study.

**H7.** *Information sharing has a significant mediating effect between digital transformation and sustainable supply chain performance.*

### 3.1.8. Mediating Role of Traceability

Digital transformation strengthens the dynamic capabilities of the supply chain. Through traceability technology, this manifests itself on the one hand in a dynamic ability to anticipate threats and on the other hand in an increase in the operational efficiency of the supply chain [15]. During operational management, traceability systems monitor the quality and safety of medicines, enable redundancy in inventory management and obtain accurate tracking and location [10]. For the supply chain, it effectively enforces accountability, reduces procurement time and improves transport efficiency. In terms of performance, it can combat counterfeit medicines, reduce consumption and shrink costs. Digital transformation as a driving force for traceability assists the supply chain to remove barriers and grow steadily [85,86]. Therefore, the following hypothesis is proposed in this study.

**H8.** *Traceability has a significant mediating effect between digital transformation and sustainable supply chain performance.*

### 3.1.9. Mediating Role of Information Sharing and Traceability

One of the goals of the accelerated digital transformation in healthcare is to achieve sustainable social, economic and environmental performance. Kim & Lee 2021 confirmed this view using an empirical study [87]. As the pharmaceutical supply chain is a matter of life and health, governments are trying to improve safety regulation through digital technologies. Aldawood et al. 2019 review cybersecurity in health and find that the use of digital technologies can restructure healthcare services and revolutionize healthcare models, such as telemedicine, online pharmacies, etc. The sustainable and stable development of social healthcare system will be achieved through rational allocation of healthcare resources. In order for the healthcare supply chain to operate optimally, a large amount of data generated in the process of healthcare delivery will be fully extracted by the relevant departments of the supply chain in a shared manner. However, sensitive medical

information involving patients' privacy must be secured using a regulatory system with traceability and accountability [88]. In recent years, there has been a lot of research into the application of blockchain technology in the pharmaceutical supply sector. Al-Farsi et al. 2021 proposed that in terms of enterprise economic performance, protocol data such as production planning, procurement, manufacturing, delivery and returns of pharmaceutical products, mostly involve commercially sensitive information. Blockchain-integrated supply chain solutions can effectively protect data information in the supply system from leakage and tampering during sharing [89]. Centobelli et al. 2022 also confirmed through their study that blockchain, with its technical features, it builds trust relationships for supply chain members and improves the transparency of the supply chain through traceability, contributing to the development of a virtuous circular economy [90]. Roy 2021 also highlighted in a study comparing supply chain traceability and visibility that, supply chain traceability, a key element of effective supply management, is centered on information sharing, both of which are considered to be prerequisites for achieving supply chain performance [78]. Therefore, the following hypothesis is proposed in this study.

**H9.** *Information sharing and traceability have a significant mediating effect between digital transformation and sustainable supply chain performance.*

Based on the above research hypotheses, this study introduces the distal mediation concept as Figure 2.

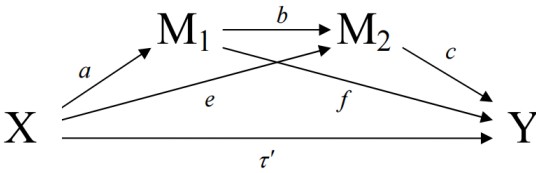

**Figure 2.** Distal mediation model.

The mediation model includes two mediators in a causal sequence. The antecedent (X) on the consequence (Y) as it is mediated first by one mediator (M1) and then another mediator (M2), (i.e., X → M1 → M2 → Y). This process is referred to as distal mediation [91,92]. The research model was constructed based on the above research hypotheses as Figure 3.

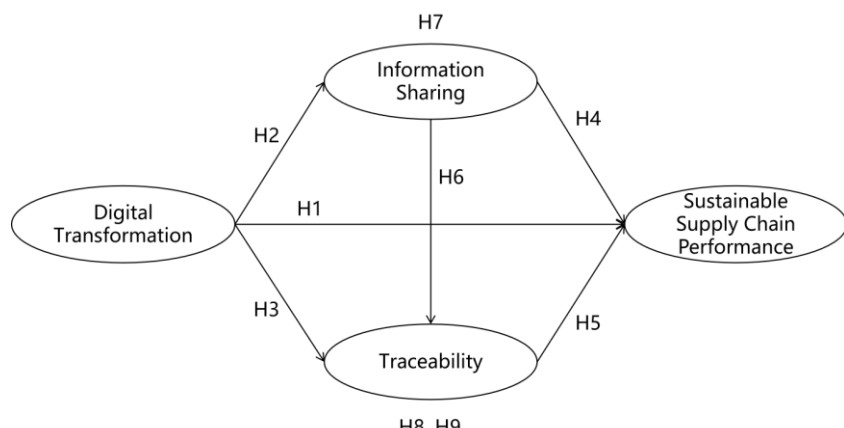

**Figure 3.** Research Model.

### 3.2. Measurement

The questionnaire was designed based on the research of many of the aforementioned scholars. The variables measured in the study were extracted from established research scales. Five questions were set for each variable, including the independent variable Digital Transformation, the mediator variable Information Sharing and Traceability, and the dependent variable Sustainable Supply Chain Performance. In this study, the Likert

5-point scale was chosen to measure the following: strongly disagree, disagree, average, agree and strongly agree, with each response being scored 1, 2, 3, 4 and 5 in that order. The details of the items asked are shown in Table 1.

**Table 1.** Variable Measurement.

| Variables | | Items | Sources |
|---|---|---|---|
| Digital Transformation | DT1 | Digital transformation enhance information systems capabilities. | Nasiri et al., (2020) [93] Singhdong et al., (2021) [94] |
| | DT2 | Digital transformation create networks between different businesses. | |
| | DT3 | Digital transformation allows for the collection of large amounts of data from different sources. | |
| | DT4 | Digital transformation aligns business and information systems. | |
| | DT5 | Digital transformation enhance efficient customer interfaces. | |
| Information Sharing | IS1 | Supply chain uses information sharing to share information with partners in a timely manner | Yatuwa, (2020) [74] Kim et al., (2021) [87] |
| | IS2 | Supply chains use information sharing to respond to the mutual needs of supply chain members timely. | |
| | IS3 | Supply chain uses information sharing to minimize distortion of communication information. | |
| | IS4 | Supply chain can grasp market demand by using information sharing. | |
| | IS5 | Supply chains use information sharing to share strategic direction with partners. | |
| Traceability | T1 | Supply chain traceability facilitates identification of drug location information. | Kim et al., (2020) [86] Kiswili et al., (2021) [95] Zhang et al., (2020) [96] |
| | T2 | Supply chain traceability for tracking the status of drug shipments. | |
| | T3 | Supply chain traceability facilitates the tracking of drugs from raw material to endpoint. | |
| | T4 | Supply chain traceability facilitates tracking the source of drug quality problems. | |
| | T5 | Supply chain traceability facilitates recall of drugs with safety concerns. | |
| Sustainable Supply Chain Performance | SSCP1 | Sustainable supply performance maximizes profits. | Kim et al., (2020) [86] Lee et al., (2021) [76] Owago et al., (2021) [85] Lu et al., (2019) [77] |
| | SSCP2 | Sustainable supply performance rationalizes total product cost. | |
| | SSCP3 | Sustainable supply performance leads to sustained improvement in financial performance. | |
| | SSCP4 | Sustainable supply performance leads to increased production profitability. | |
| | SSCP5 | Sustainable supply performance can improve supply chain vulnerability. | |

### 3.3. Demographics

The respondents of this study are professional supply chain management personnel of pharmaceutical companies, and the managers should have one year or more working experience in this field with a bachelor's degree or above. The geographical scope of the survey was the main provinces and cities of Chinese pharmaceutical companies. 329 questionnaires were collected, 31 invalid questionnaires with logically contradictory answers were removed, and 298 valid questionnaires were collected in total. The study used SPSS

26.0 and AMOS 24.0 to empirically analysis the sample data. The demographic characteristics are shown in Table 2. The respondents were predominantly young and middle-aged people under the age of 40, with 44.2% of the age group under 30 years old and 50.0% of the age group 30–40 years old. The most highly educated group mainly holds bachelor's degrees, accounting for 82.2%. The percentage of those with 1–5 years of experience in pharmaceutical companies is 71.4%. In China's "Regulations on the Classification of Small and Medium-sized Enterprises", in the industrial sector, those with an annual business revenue of less than 2 billion RMB are considered small, medium and micro enterprises [97]. The annual sales (RMB) of the pharmaceutical companies to which they belonged were 200 million (54.3%) and 200 million to 2 billion (31.5%), indicating that the respondents were mostly managers of medium-sized and small pharmaceutical companies in China. The surveyed executives' main areas of work are first-tier, second-tier and third-tier cities in China, with large and medium-sized cities predominating. This is in line with the geographical distribution of the pharmaceutical market in China.

**Table 2.** Demographic Characteristics of the Respondents.

| Variables | Category | Frequency | Ratio (%) |
|---|---|---|---|
| Gender | Male | 152 | 51.0 |
| | Female | 146 | 49.0 |
| Age | <30 | 132 | 44.3 |
| | 30–40 | 149 | 50.0 |
| | 41–50 | 14 | 4.7 |
| | >50 | 3 | 1.0 |
| Education Background | Bachelor | 245 | 82.2 |
| | Master | 46 | 15.5 |
| | Doctor | 7 | 2.3 |
| work seniority | 1–5 | 213 | 71.5 |
| | 5–10 | 64 | 21.5 |
| | >10 | 21 | 7.0 |
| Annual Company Turnover (million RMB) | <200 | 162 | 54.4 |
| | 200–2000 | 94 | 31.5 |
| | >2000 | 42 | 14.1 |
| Major Service District (Multiple selection) | First-tier cities | 158 | 33.5 |
| | Second-tier cities | 122 | 25.8 |
| | Third-tier cities | 96 | 20.3 |
| | Fourth-tier cities | 69 | 14.6 |
| | Fifth-tier cities | 27 | 5.7 |

## 4. Data Analysis and Results

### 4.1. Exploratory Factor Analysis

In this study, an exploratory factor analysis was conducted using SPSS 26.0 to examine the reliability and validity. The results are shown in Table 3: It is generally considered that the value of Cronbach's $\alpha$, if Cronbach's $\alpha$ is above 0.9, internal consistency is excellent; if Cronbach's $\alpha$ is above than or 0.8, internal consistency is good; if Cronbach's $\alpha$ is above 0.7, internal consistency is acceptable; if Cronbach's $\alpha$ is above 0.6, internal consistency is questionable; if Cronbach's $\alpha$ is above 0.5, internal consistency is poor; if Cronbach's $\alpha$ is below 0.5, internal consistency would be unacceptable [98]. In the KMO test, if KMO is above 0.9, it is marvelous; if KMO is above 0.8, it is meritorious; if KMO is above 0.7, it is middling; if KMO is above 0.6, it is mediocre; if KMO is above 0.5, it is miserable; if KMO is below 0.5, it would be unacceptable [99]. The KMO value for this study was 0.955 and the data was well suited for factor analysis. Bartlett's spherical test was 7051.682 (df = 190) with sig = 0.000, which met the significance index of $p < 0.05$ and the data was taken from a normal distribution and was suitable for factor analysis. The cumulative variance explained was 82.488%, which met the required criterion of 50% or more. The absolute values of the

factor loading coefficients were all greater than 0.6, and the convergent validity between each question item and the corresponding factor was good. The results of the analysis showed that the reliability and validity of the model data met the criteria and were stable and reliable.

**Table 3.** Exploratory Factor Analysis Results.

| Variables | Codes | Factor Loading | | | | Cronbach's α |
|---|---|---|---|---|---|---|
| | | **1** | **2** | **3** | **4** | |
| Digital Transformation | DT1 | 0.293 | 0.168 | 0.323 | 0.815 | |
| | DT2 | 0.318 | 0.222 | 0.285 | 0.804 | |
| | DT3 | 0.308 | 0.296 | 0.290 | 0.705 | 0.937 |
| | DT4 | 0.360 | 0.335 | 0.245 | 0.683 | |
| | DT5 | 0.462 | 0.381 | 0.161 | 0.627 | |
| Information Sharing | IS1 | 0.213 | 0.263 | 0.796 | 0.249 | |
| | IS2 | 0.279 | 0.333 | 0.667 | 0.328 | |
| | IS3 | 0.261 | 0.403 | 0.655 | 0.299 | 0.923 |
| | IS4 | 0.277 | 0.401 | 0.686 | 0.271 | |
| | IS5 | 0.291 | 0.300 | 0.742 | 0.206 | |
| Traceability | T1 | 0.314 | 0.707 | 0.346 | 0.279 | |
| | T2 | 0.341 | 0.634 | 0.435 | 0.263 | |
| | T3 | 0.272 | 0.771 | 0.319 | 0.194 | 0.936 |
| | T4 | 0.266 | 0.709 | 0.414 | 0.275 | |
| | T5 | 0.261 | 0.748 | 0.295 | 0.307 | |
| Sustainable Supply Chain Performance | SSCP1 | 0.826 | 0.268 | 0.272 | 0.322 | |
| | SSCP2 | 0.809 | 0.255 | 0.301 | 0.280 | |
| | SSCP3 | 0.818 | 0.255 | 0.263 | 0.328 | 0.976 |
| | SSCP4 | 0.822 | 0.272 | 0.247 | 0.288 | |
| | SSCP5 | 0.842 | 0.283 | 0.237 | 0.316 | |
| Eigen Value (Rotated) | | 4.796 | 3.930 | 3.903 | 3.869 | |
| Explained Variance (%) | | 23.981 | 19.648 | 19.516 | 19.343 | |
| Cumulative Variance (%) | | 23.981 | 43.629 | 63.145 | 82.488 | |
| KMO = 0.955, Bartlett = 7051.682, Sig = 0.000, df = 190 | | | | | | |

*4.2. Confirmatory Factor Analysis*

In this study, validation factor analysis was conducted using AMOS 24.0 to further validate the reliability and validity of the scale data. The results are shown in Table 4. The Combined Reliability (C.R.) values for each factor in this study were all above 0.9, meeting the general criterion of above 0.7. The Average Variance Extracted (AVE) values were all above 0.7, meeting the general criterion of above 0.5, and the data had good convergent validity. The Chi-square/DF value was 3.221, close to the ideal value of 3. GFI = 0.848 and AGFI = 0.806, both above the standard value of 0.8. TLI = 0.940 and CFI = 0.948, both above the ideal value of 0.9. The scale data are consistent with the model fitness index [100].

**Table 4.** Confirmatory Factor Analysis Results.

| Variables | Codes | Unstd. | S.E. | T-Value | $p$ | Std. | C.R. | AVE |
|---|---|---|---|---|---|---|---|---|
| | DT1 | 1 | | | | 0.906 | | |
| | DT2 | 1.014 | 0.039 | 25.688 | *** | 0.918 | | |
| Digital Transformation | DT3 | 0.981 | 0.048 | 20.621 | *** | 0.837 | 0.938 | 0.752 |
| | DT4 | 0.983 | 0.048 | 20.679 | *** | 0.838 | | |
| | DT5 | 0.953 | 0.047 | 20.443 | *** | 0.833 | | |
| | IS1 | 1 | | | | 0.827 | | |
| | IS2 | 0.948 | 0.054 | 17.523 | *** | 0.837 | | |
| Information Sharing | IS3 | 0.945 | 0.053 | 17.951 | *** | 0.850 | 0.923 | 0.705 |
| | IS4 | 0.993 | 0.053 | 18.606 | *** | 0.869 | | |
| | IS5 | 0.998 | 0.059 | 16.825 | *** | 0.815 | | |
| | T1 | 1 | | | | 0.869 | | |
| | T2 | 0.968 | 0.047 | 20.658 | *** | 0.870 | | |
| Traceability | T3 | 1.076 | 0.056 | 19.273 | *** | 0.839 | 0.937 | 0.747 |
| | T4 | 1.015 | 0.047 | 21.598 | *** | 0.890 | | |
| | T5 | 0.983 | 0.049 | 19.905 | *** | 0.854 | | |
| | SSCP1 | 1 | | | | 0.965 | | |
| | SSCP2 | 0.96 | 0.029 | 33.139 | *** | 0.917 | | |
| Sustainable Supply Chain Performance | SSCP3 | 0.988 | 0.024 | 40.365 | *** | 0.952 | 0.976 | 0.892 |
| | SSCP4 | 0.972 | 0.029 | 33.684 | *** | 0.920 | | |
| | SSCP5 | 1.013 | 0.023 | 44.705 | *** | 0.967 | | |

CMIN = 528.178, DF = 164, CMIN/DF = 3.221, GFI = 0.848, AGFI = 0.806, TLI = 0.940, CFI = 0.948, RMSEA = 0.086, SRMR = 0.0343

Note: *** $p < 0.000$.

### 4.3. Correlation Analysis

This study used SPSS 26.0 to analyze the degree of correlation between the factors through the Pearson method. The results of the analysis are shown in Table 5. The correlations between the factors were all significant. The absolute magnitude of the observed correlation coefficient, if correlation coefficient is above 0.9, it means very strong correlation; if it is above 0.7, it means strong correlation; if it is above 0.40, it means moderate correlation; if it is above 0.10, it means weak correlation. [101]. The results of this study show that the correlation coefficient values for Digital transformation, information sharing, traceability, and sustainable supply chain performance fluctuate from 0.690 to 0.822 and are all significant ($p < 0.000$). This indicates that there is a strong correlation between the factors. The mean value of the variables fluctuates from 4.4 to 4.476. The maximum standard deviation is 0.602 and the minimum is 0.557, indicating that the distribution of the variables is relatively concentrated and dispersion is rare.

**Table 5.** Correlation Analysis Results.

| Variables | M | SD | 1 | 2 | 3 | 4 |
|---|---|---|---|---|---|---|
| Digital Transformation | 4.476 | 0.557 | 1 | | | |
| Information Sharing | 4.396 | 0.593 | 0.728 ** | 1 | | |
| Traceability | 4.443 | 0.601 | 0.733 ** | 0.822 ** | 1 | |
| Sustainable Supply Chain Performance | 4.462 | 0.602 | 0.763 ** | 0.690 ** | 0.709 ** | 1 |

Note: ** $p < 0.01$.

### 4.4. Path Analysis

In this study, path analysis was performed on the sample data using AMOS 24.0. The results are shown in Table 6. The path coefficient of digital transformation on sustainable supply chain performance is 0.546 ($p = 0.000$) significant, and the hypothesis H1 is supported. The path coefficient of digital transformation on information sharing is 0.825 ($p = 0.000$) significant, and the hypothesis H2 is supported. The path coefficient of digital transformation on traceability is 0.195 ($p = 0.001$) significant, and hypothesis H3

is supported. The path coefficient of information sharing on sustainable supply chain performance is 0.082 ($p$ = 0.457) is not significant and hypothesis H4 is not supported. The path coefficient of traceability on sustainable supply chain performance is 0.305 ($p$ = 0.005) significant and hypothesis H5 is supported. The path coefficient of information sharing on traceability is 0.739 ($p$ = 0.000) significant, hypothesis H6 is supported.

**Table 6.** Path Analysis Results.

| Hypothesis | Path | Standardized Estimate | SE | CR | $p$ | Results |
|---|---|---|---|---|---|---|
| H1 | DT→SSCP | 0.546 | 0.073 | 7.439 | *** | Supported |
| H2 | DT→IS | 0.825 | 0.059 | 14.036 | *** | Supported |
| H3 | DT→T | 0.195 | 0.061 | 3.180 | 0.001 | Supported |
| H4 | IS→SSCP | 0.082 | 0.111 | 0.744 | 0.457 | Not Supported |
| H5 | T→SSCP | 0.305 | 0.108 | 2.809 | 0.005 | Supported |
| H6 | IS→T | 0.739 | 0.068 | 10.856 | *** | Supported |
| CMIN = 528.178, DF = 164, CMIN/DF = 3.221, GFI = 0.848, AGFI = 0.806, TLI = 0.940, CFI = 0.948, RMSEA = 0.086, SRMR = 0.0343 | | | | | | |

Note: *** $p$ < 0.000.

### 4.5. Test of Mediating Effect

In this study, the test of mediating effect was performed by bootstrapping (5000) using AMOS 24.0. The results are shown in Table 7. The point estimate of the total effect of digital transformation on sustainable supply chain performance through mediation is 0.859 with a standard error of 0.070. The Bias-Corrected 95% confidence interval for lower bound is 0.742 and upper bound is 1.023, zero is not between the lower and upper bound, $p$ = 0. 000, the mediating effect existed. Direct effect shows point estimate is 0.546, standard error is 0.147. Lower bound is 0.263, upper bound is 0.847, zero is not between lower and upper bound, $p$ = 0.000, the direct effect existed. Indirect effect results show a point estimate of 0.313 with a standard error of 0.122. Lower bound is 0.106, upper bound is 0.580, zero is not between the lower and upper bound, $p$ = 0.003, the indirect effect existed. Thus information sharing and traceability between digital transformation and sustainable supply chain performance play a partially mediating role [102,103].

**Table 7.** Mediating Effect Test Results.

| Mediating | Hypothesis | Point Estimation | SE | Bootstrapping Bias-Corrected 95% CI | | | Results |
|---|---|---|---|---|---|---|---|
| | | | | Lower | Upper | $p$ | |
| | | | | Total effect | | | |
| | - | 0.859 | 0.070 | 0.742 | 1.023 | 0.003 | - |
| DT→SSCP | | | | Direct effect | | | |
| | - | 0.546 | 0.147 | 0.263 | 0.847 | 0.000 | - |
| | | | | Indirect effect | | | |
| | - | 0.313 | 0.122 | 0.106 | 0.580 | 0.003 | - |
| DT→IS→SSCP | H7 | 0.068 | 0.153 | −0.257 | 0.358 | 0.706 | Not Supported |
| DT→T→SSCP | H8 | 0.059 | 0.049 | 0.001 | 0.219 | 0.045 | Supported |
| DT→IS→T→SSCP | H9 | 0.181 | 0.108 | −0.004 | 0.436 | 0.054 | Not Supported |

Note: 5000 bootstrap samples.

Based on the fact that the total effect was significant and the indirect effect was significant, we continued to investigate the significance of each of the mediating variables information sharing and traceability, and whether the remote mediation consisting of information sharing to traceability was significant. The results of the study showed that the mediating effect of information sharing in the middle of digital transformation and Sustainable Supply Chain Performance, lower bound is −0.257 and upper bound is 0.358, zero is between the lower and upper bound, $p$ = 0.706, which is not significant and hy-

pothesis H7 is not supported. The mediating effect of traceability in the middle of digital transformation and sustainable supply chain performance, lower bound is 0.001, upper bound is 0.219, zero is not between the lower and upper bound, $p = 0.045$, hypothesis H8 is supported. Digital transformation affects sustainable supply chain performance by the distal mediation of information sharing and traceability, the result is that lower bound is $-0.004$, upper bound is 0.436, zero is between the lower and upper bound, $p = 0.054$, insignificant, hypothesis H9 is not supported.

## 5. Discussion and Conclusions

### 5.1. Discussion

With the increasing demand in the pharmaceutical market and in the era of rapid development of information technology, the pharmaceutical supply chain needs to achieve continuous growth in performance through efficient management mechanisms to assist the supply chain. This paper provides a comprehensive analysis of the relationship between the four components of digital transformation, information sharing, traceability and sustainable supply chain performance. The results of the empirical analysis reveal that digital transformation is beneficial to the pharmaceutical supply chain in achieving sustainable supply performance. Digital transformation has a significant direct impact on both information sharing and traceability. This is in line with the findings of Alabdali & Salam 2022, Massaro 2021, Omar et al. 2022 [61,69,72]. Therefore, enhanced digital transformation in the pharmaceutical supply chain contributes to the ability to manage information sharing and traceability, which is conducive to achieving sustainable development in the pharmaceutical supply sector. Traceability in supply chain management directly affects sustainable supply performance, and traceability as a mediator can enhance the positive impact of digital transformation on sustainable supply performance, corroborating the findings of Lee & Ha 2021, Haji et al. 2021 [10,76]. Traceability as an important driver can therefore assist the pharmaceutical supply chain in achieving superior performance through digital transformation. The direct effect of information sharing on sustainable supply performance was not significant. This is contrary to the results of Guan et al. 2020, Yatuwa 2020 study [73,74]. The mediating effect of information sharing was not significant during the test of mediating effects of digital transformation and supply performance. This is different from the findings of Nestle et al. 2019, Li 2022 [64,84]. Besides, the distal effect of consisting of information sharing and traceability remains insignificant. However, the total effect of the mediating effect is significant, indicating that information sharing and traceability as two independent trends can have synergistic effects. Due to the uncoordinated development of digital technologies among supply chain members, technological uncertainty will affect the quality of information exchanged between partners, which in turn affects trust between stakeholders and has a negative impact on ultimate performance. Xu et al. (2022) found that technological uncertainty has a U-shaped relationship with inter-organizational solidarity trust in the digital transformation of supply chain management through an empirical study [34].

In addition, this study is compared with Kamble et al. 2021's study on the impact of blockchain technology on sustainable supply chain performance for the automotive industry. Kamble et al. 2021's study in the automotive sector has some commonalities with my study in the pharmaceutical supply sector. However, Kamble et al. 2021 validated the role of supply chain integration as a mediator in their study. Supply chain integration involves inter-organizational collaboration, organizational forecasting of demand, and adaptation to demand fluctuations [104]. This study visualizes supply chain integration capabilities to provide clearer guidance to firms for development by examining the mediating role of information sharing and traceability. Therefore, this study may have the same theoretical contribution in the field of automotive supply chain.

*5.2. Conclusions*

5.2.1. Theoretical Implications

This study confirms the positive impact of digital transformation on sustainable supply chain performance. The management of the pharmaceutical supply chain is pertinent to the theory. Consequently, the domain in which the theory is expanded. The results of the empirical study with Chinese pharmaceutical supply chain managers as respondents also demonstrate the applicability of the theory to the geographical scenario in China. By comparing the effects of the media of information sharing and traceability, it fills in the research gaps of related theories in pharmaceutical supply management. It enriches the theoretical research related to the information sharing and traceability components of remote media.

5.2.2. Practical Implications

First, empirical study results show digital transformation can improve sustainable supply chain performance. The study's findings suggest that as digital construction rapidly revolutionizes the healthcare sector, pharmaceutical companies must keep pace, and cutting-edge pharmaceutical companies have now accelerated the deployment of blockchain, internet of things, big data and other technologies. In the future, precision medicine on an individual basis will dominate the healthcare market, and the existing homogeneous mass production will shift to a small batch, flexible, precise and efficient production model. Supply chain management systems will be required with efficient collaboration capabilities and lean traceability. So that pharmaceutical companies in the complex and changing and competitive pharmaceutical supply environment to successfully complete the transformation.

Second, the results of the mediating role show that the traceability of pharmaceutical supply management plays a mediating role to facilitate the positive impact of digital transformation on supply chain performance. The results of this study reveal that society has become more stringent in regulating drug safety due to the increased importance people place on life and health. The supply management of pharmaceutical companies must have a strong safety monitoring system to ensure production safety, transportation safety and operational safety. Traceability technology, represented by blockchain, is fully developed and applied for accountability in safety management, which makes the overall pharmaceutical supply chain environment clearer.

Third, this study focuses on the sustainability of pharmaceutical supply chains in the theoretical examination. The results of the empirical study are combined with insights on the improvement of the economic performance and vulnerability of the supply chain. emphasis on sustainable development: The risk management built by the network system is no longer confined to monitoring and response, but has shifted to continuously gaining energy from external shocks and developing in a proactive and reinforcing way of growth. The company's long-term strategic objectives are also not limited to financial performance, in the steady self-growth to take competitive advantage needs to take into account the society and nature to obtain energy, and will benefit back to the market economy, social civilization and environmental protection, to achieve long-term sustainable and healthy development.

5.2.3. Limitations

The survey respondents were only restricted to managers of pharmaceutical enterprises in the pharmaceutical supply chain. Future research can narrow the geographical scope of the survey respondents, compare the degree of digital transformation of pharmaceutical enterprises among local provinces in China, and find out whether information interoperability and sharing is hindered by the existence of digital technology construction gaps, so as to break the barriers of uneven development. Additionally, in the future, it can include the attitude of healthcare providers towards digital transformation of the pharmaceutical supply chain to achieve comprehensive and sustainable development of the industry chain.

**Author Contributions:** Writing—original draft preparation and writing—review and editing, J.-Y.M.; Conceptualization and data analysis, T.-W.K.; investigation, L.S. All authors have read and agreed to the published version of the manuscript.

**Funding:** This research received no external funding.

**Institutional Review Board Statement:** The study was conducted according to the guidelines of the Declaration of Helsinki, and approved by Kunsan University Bioethics Committee (1040117-202211HR-032-02, 2 December 2022).

**Informed Consent Statement:** Informed consent was obtained from all subjects involved in the study.

**Data Availability Statement:** The datasets of this study are available from the corresponding author on reasonable request.

**Conflicts of Interest:** The authors declare that they have no known competing financial interests or personal relationships that could have influenced the work reported in this paper. There are no conflict of interest.

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
