# Peer review of "The Effect of Digital Transformation on the Pharmaceutical Sustainable Supply Chain Performance: The Mediating Role of Information Sharing and Traceability Using Structural Equation Modeling"

_sustainability, doi:10.3390/su15010649_

Round 1

Reviewer 1 Report

Happy to review your article. Please address the following:

1.       Please rephrase “in the market dividend period” in line 31.

2.       Check the abbreviation for “information physical systems” (CPS) in line 42.

3.       Blockchain is one-word revise in the whole document.

4.       Explain the concept of traceability in introduction section.  

5.       Explain the concept of distal mediation in hypothesis section.

6.       Lack of support for H9 needs to be discussed in more detail looking at the fact that information sharing is essential for traceability.

7.       Conclusion needs to connect better with the outcome of the study.

Author Response

Dear Professor

Thank you very much for the valuable comments you have provided. It helps to improve my research. I have carefully revised it according to your suggestions. Please see the attachment.

Once again, we express our sincere thanks.

Best regards,
Dr. Ma

Reviewer 2 Report

Thank you for the opportunity to review your manuscript. 

Summary: The authors studied the impact of digital transformation on sustainable supply chain performance and the role of information sharing and traceability as a medium. The topic is interesting and timely given the increasing interest in establishing end-to-end supply chain traceability in the pharmaceutical sector. My comments on the different sections of the paper are provided below. 

The title: It reflects the aim and the context of the paper but not the method applied

The abstract: While the abstract is nicely written, it is not informative. The authors need to provide the general background of the work, highlight the knowledge gap, and the theoretical contributions of the paper. 

The introduction: 

L30: Can you support your statement with statistics? 

L43: CPS stands for cyber-physical systems and not information physical systems. Please correct. 

The introduction flows nicely, and it presents the general background of the paper. To improve this section, the authors should highlight the knowledge gap by briefly reviewing the prior empirical studies on the topic. This not only helps to justify the need for the current investigation but also clarifies the novelty of the work. Moreover, the authors should indicate how they will address the knowledge gap by presenting their research method. L82- L84. Comparing your study to these works is misleading because both studies are generic and do not fall within the pharmaceutical sector. I suggest you reference prior empirical studies in the pharmaceutical sector, or at least in different sectors, to differentiate the impact of digital transformation on the supply chain of other sectors. 

A few sentences on the structure of the paper should be inserted. 

L126-L135: Please support your statements with the relevant literature. Some suggestions for your consideration: 

https://www.mdpi.com/2305-6290/4/4/27

https://www.igi-global.com/chapter/content/72194

https://www.tandfonline.com/doi/abs/10.1080/00207543.2018.1533261

Please fix the inconsistencies in the formatting of the references 

L148- L154: Please support your statement with the literature 

L159-l66: Please support your statement with the relevant literature 

L168-L171: Please support your sentence with the relevant literature. For example, you can consider the following study 

https://link.springer.com/article/10.1007/s42488-021-00046-2

L174-176: Please support your statement with the relevant literature

L179: Please add a reference to support your statement 

L180-182: Can you elaborate on the TBL concept? 

L187: You provided a definition without supporting it from the relevant literature 

L190: Please include a brief description of these concepts in a table supporting it with the relevant literature. 

L193-194: Unsupported claim. How can digital networking help supply chains achieve a competitive advantage? 

L194-L198: Again, unsupported claim. Please support your statements from the literature. 

L201-L205: redundant sentences

L212-222: Unsupported claims 

L234-248: Unsupported and baseless claims

L253-L259: Please reference the relevant literature to support your arguments. 

L263: a grammar mistake. 

L282-L290: Baseless arguments 

L296-L311: Unsupported arguments 

L316-325: Unsupported section. 

L332-L339: Same issue

L346-L355: Again, you did not refer to the relevant literature to support your arguments 

L361: Please remove the heading.

In the discussion section, I suggest comparing the findings of the authors to the findings of other scholars investigating the same or a closely related topic in another sector. I am curious to know if these findings apply only to the pharmaceutical industry or may go beyond. 

The conclusion lacks more rigor. Please answer your research questions briefly and divide the conclusion into three sections: 

Theoretical implications 

Practical implications 

Limitations 

Overall, the authors did a good job, but the major weakness of the paper is the high number of unsupported claims and arguments. You should have based your research on the relevant literature, otherwise, your arguments will simply be nothing more than viewpoints and opinions. 

Good luck with your revisions

Author Response

(The authors gave the same response as above.)

Reviewer 3 Report

The description of solution method needs to be added in the abstract. 

The paper needs to have literature review section.

The literature review should assess strengths and weaknesses of individual studies. Also, it should assess the existing research as a whole. 

The literature should identify potential gaps in knowledge.

The literature should outline important research trends and it should establish a need for current and/or future research projects.

The presented solution method by the authors needs to be compared your other solution methods in the literature. 

In the conclusion you need to discuss how your research findings is different from other papers in the field of study. 

Author Response

(The authors gave the same response as above.)

Round 2

Reviewer 2 Report

The authors answered all comments properly. Therefore, I recommend the publication of this manuscript in its current form. 

Reviewer 3 Report

Accept in present form